# Targeting prolyl-tRNA synthetase via a series of ATP-mimetics to accelerate drug discovery against toxoplasmosis

**Manickam Yogavel**[1☯], **Alexandre Bougdour**[2☯], **Siddhartha Mishra**[1,3,4☯], **Nipun Malhotra**[1☯¤], **Jyoti Chhibber-Goel**[1], **Valeria Bellini**[2], **Karl Harlos**[5], **Benoît Laleu**[6], **Mohamed-Ali Hakimi**[2]*, **Amit Sharma**[1,3,4]*

**1** Molecular Medicine–Structural Parasitology Group, International Centre for Genetic Engineering and Biotechnology (ICGEB), Aruna Asaf Ali Marg, New Delhi, India, **2** Institute for Advanced Biosciences (IAB), Team Host-Pathogen Interactions and Immunity to Infection, INSERM U1209, CNRS UMR5309, Université Grenoble Alpes, Grenoble, France, **3** Academy of Scientific and Innovative Research (AcSIR), Ghaziabad, India, **4** ICMR-National Institute of Malaria Research, Dwarka, New Delhi, India, **5** Division of Structural Biology, Wellcome Centre for Human Genetics, University of Oxford, Oxford, United Kingdom, **6** Medicines for Malaria Venture (MMV), International Center Cointrin (ICC), Geneva, Switzerland

☯ These authors contributed equally to this work.
¤ Current address: David Heart and Lung Institute, Wexner Medical Center, The Ohio State University, Columbus, United States of America
* mohamed-ali.hakimi@univ-grenoble-alpes.fr (M-AH); amit.icgeb@gmail.com, directornimr@gmail.com, amitpsharma68@gmail.com (AS)

**Data Availability Statement:** All relevant data are within the manuscript and its Supporting information files. The data that are freely accessible now include x-ray macromolecular protein-

## Abstract

The prolyl-tRNA synthetase (PRS) is a validated drug target for febrifugine and its synthetic analog halofuginone (HFG) against multiple apicomplexan parasites including *Plasmodium falciparum* and *Toxoplasma gondii*. Here, a novel ATP-mimetic centered on 1-(pyridin-4-yl) pyrrolidin-2-one (PPL) scaffold has been validated to bind to *Toxoplasma gondii* PRS and kill toxoplasma parasites. PPL series exhibited potent inhibition at the cellular (*T. gondii* parasites) and enzymatic (*Tg*PRS) levels compared to the human counterparts. Cell-based chemical mutagenesis was employed to determine the mechanism of action via a forward genetic screen. *Tg*-resistant parasites were analyzed with wild-type strain by RNA-seq to identify mutations in the coding sequence conferring drug resistance by computational analysis of variants. DNA sequencing established two mutations, T477A and T592S, proximal to terminals of the PPL scaffold and not directly in the ATP, tRNA, or L-pro sites, as supported by the structural data from high-resolution crystal structures of drug-bound enzyme complexes. These data provide an avenue for structure-based activity enhancement of this chemical series as anti-infectives.

## Author summary

Nearly one-third of the global population is chronically infected with the apicomplexan parasite *Toxoplasma gondii*. It does not particularly have any drastic impacts on a healthy individual with a robust immune response. But in immune-compromised patients, the

compound structures accessible at the RCSB-PDB website. These include the following with their DOIs • 7EVV – 10.2210/pdb7EVV/pdb • 7VC1 – 10.2210/pdb7VC1/pdb • 7FAK – 10.2210/pdb7FAK/pdb • 7VC2 – 10.2210/pdb7VC2/pdb • 7FAM – 10.2210/pdb7FAM/pdb • 7VC3 – 10.2210/pdb7VC3/pdb • 7FAN – 10.2210/pdb7FAN/pdb • 7FAL – 10.2210/pdb7FAL/pdb • 7F9B – 10.2210/pdb7F98/pdb • 7F98 – 10.2210/pdb7F98/pdb • 7F9C – 10.2210/pdb7F9C/pdb • 7F99 – 10.2210/pdb7F99/pdb • 7F9D – 10.2210/pdb7F9D/pdb • 7F9A – 10.2210/pdb7F9A/pdb Additionally, The Illumina RNA-seq dataset generated during this study has been deposited at the National Center for Biotechnology Information Gene Expression Omnibus (NCBI GEO) under accession number GSE217090 (https://www.ncbi.nlm.nih.gov/geo/query/acc.cgi?acc=GSE217090).

**Funding:** The work was supported by the Medicines for Malaria Venture (MMV) Project grant P020/00065 and grant from the Department of Biotechnology (DBT), Government of India (PR32713). KH is supported by the UK Medical Research Council (grant MR/V001329/1) and the Wellcome Centre for Human Genetics (grant 090532/Z/09/Z). J. C. Bose National Fellowship (SB/S2/JCB-41/2013) from the Department of Science and Technology (DST) supports AS. JCG is funded by the Department of Biotechnology, under the BioCARe scheme (BT/PR30603/BIC/101/1104/2018). M-A.H, A.B and V.B. were funded by the Laboratoire d'Excellence (LabEx) ParaFrap [ANR-11-LABX-0024], the Agence Nationale pour la Recherche (Project HostQuest, ANR-18-CE15-0023, Project ApiNewDrug, ANR-21-CE35-0010-01, Project ToxoP53, ANR-19-CE15-0026-01), and Fondation pour la Recherche Médicale (FRM Equipe # EQU202103012571). The funders had no role in study design, data collection and analysis, decision to publish, or preparation of the manuscript.

**Competing interests:** We have read the journal's policy and the authors of this manuscript declare the competing interest that BL is an MMV employee.

parasite has been found to wreak havoc. The necessity of developing novel antibiotic therapeutics against this ailment is reprised by the limited efficacy of the SP regimen generally prescribed to patients with toxoplasmosis–including suppression of active proliferation–but not the latent stage. This regime has its limitations in the clearance of chronic infection and in the fact that crossing over the blood-brain barrier is troublesome for small molecules, which might not be able to prevent ocular toxoplasmosis in severe cases. Targeting of molecular motors within the protein translation machinery such as prolyl tRNA synthetases with structure-based designed small molecules mimicking the natural substrate, ATP, forms the basis of the work published here as being an avenue for targeting the parasite selectively.

## Introduction

Aminoacyl-tRNA synthetases (aaRSs) are an essential enzyme family for protein translation and, in several cases, have been validated drug targets [1–4]. Based on their catalytic mechanisms and three-dimensional structures, aaRSs are known to possess three sub-sites of– 1) ATP, 2) amino acid, and 3) tRNA binding pocket, in addition to other subsites [3,5]. Several parasite aaRSs are currently being investigated as potential drug targets [5–21]. Parasite species targeted include obligate intracellular apicomplexans that cause malaria (*Plasmodium* spp), toxoplasmosis (*Toxoplasma gondii*), and cryptosporidiosis (*Cryptosporidium* spp) [22–25]. These parasites exhibit high conservation of housekeeping genes, and their essential invariant proteins are thus possibly reasonable targets for drug development. The limited treatment options and the emergence and spread of drug-resistant parasites highlight the need to exploit the vulnerabilities of human parasites via the discovery of new enzyme families for drug targeting.

The prolyl-tRNA synthetase (PRS) is a validated target for antimalarial drug development [26–31]. It has also been structurally and biochemically dissected in context of halofuginone (HFG) and several quinazolinone-based inhibitors (QBI) [5,8,14,26,28]. HFG and QBIs act through a proline-competitive binding mode, with more effective binding to PRS with ATP [27]. Despite the high homology between the parasite and host PRSs and comparable biochemical sensitivity, HFG is significantly more potent against *P. falciparum* at the asexual blood stage ($EC_{50}$ of $1 \pm 0.5$ nM) than the mammalian cell lines ($EC_{50}$ of $150 \pm 9$ nM) [6]. However, clinical development of HFG has been held back due to side effects and a deleterious phenotypic drug resistance mechanism (within five generations) via the accumulation of L-proline (~ 20-fold upregulation)—called the Adaptive Proline Response (APR) [32].

In 2017, Takeda Pharmaceutical Company Limited disclosed a new class of *Hs*PRS inhibitors to treat fibrosis [33]. Unlike HFG and analogs which span the A76 and proline-binding sites and thus interact in an ATP-uncompetitive manner, these new 1-(pyridin-4-yl) pyrrolidin-2-one (PPL) derivatives target the ATP-binding pocket. Some of these compounds, e.g., T-3767758, displayed proline-uncompetitive steady-state kinetics for *Hs*PRS [29]. Based on this class of inhibitors, Okaniwa et al. identified PPL derivatives as a new class of antimalarials that bind to the PRS ATP-site with double-digit nanomolar activity against *Plasmodium falciparum* (*Pf*) and *Plasmodium vivax* (*Pv*) strains [31]. In this study, five compounds centered on the PPL scaffolds were investigated for their therapeutic efficacy in killing both the *Toxoplasma* parasites and the *Toxoplasma* encoded PRS enzyme (**Fig 1**). We provide cellular, enzymatic and structural data to validate PRS as the target of this new series of ATP-mimetics that hold promise as future agents against toxoplasmosis.

**Fig 1. 2D structure.** The chemical structures of the 1-(pyridin-4-yl) pyrrolidin-2-one scaffold derivatives L95, L96, L97, L35 and L36 are shown.

## Methodology

### Compound synthesis and cellular assays

Compounds were synthesized (HPLC purity > 95%) according to reported procedures [31,33–36]. A confluent Human Foreskin Fibroblast (HFF) monolayer was infected with 2,000 tachyzoites of RH parasites expressing the NLuc luciferase (RH NLuc) for 2 h to allow parasite invasion. Each compound was added to the culture medium, and infected cells were incubated at 37° C for 48 h. To measure luminescence, the medium was replaced by 50 μL of PBS, and the luminescence assay was performed using Nano-Glo Luciferase Assay System according to the manufacturer's instructions (Promega). After 3 min of incubation, the luminescence was measured using the CLARIOstar (BMG Labtech) plate reader. $EC_{50}$ was determined using a non-linear regression analysis of normalized data assuming a sigmoidal dose-response. $EC_{50}$ values for each compound represent an average of three independent biological replicates. Statistical analyses were performed using a one-way ANOVA test with GraphPad software.

### Measurement of $CC_{50}$ for human cells and determination of selectivity index

The medium cytotoxic concentrations for mammalian cells were determined as follows. ARPE-19 and MDA-231 cell lines were plated in 96-well plates for 1 hour and incubated with exponential concentrations of the indicated compounds in a final volume of 100 μL. After 72 hours of culture, CellTiter-Blue Reagent (Promega) (20 μl per well) was added directly to each well. Plates were then incubated at 37˚C for 2 hours to allow cells to convert resazurin to resorufin before reading fluorescence [560 ± 20 nm excitation / 590 ± 10 nm emission] with the CLARIOstar (BMG Labtech) plate reader. The cytotoxicity concentrations ($CC_{50}$) for human cells were determined using a nonlinear regression curve of the normalized data. $CC_{50}$ values represent the average of at least two biological experiments. Selectivity Index (SI) was obtained by the average of the human $CC_{50}$ divided by the average of *T. gondii* $EC_{50}$. It should be noted that the EC50s were determined using a monolayer of confluent HFF host cells with reduced

metabolic activity as opposed to dividing cells. It was chosen to be done so as ARPE-19 and MDA-231 are more relevant for testing drug biological activity due to their sensitivity to drug treatment, in contrast to HFFs–which are quite robust and would overestimate the selectivity indices.

## Toxoplasma gondii random mutagenesis

Parasites were chemically mutagenized as previously described, with the following modifications [37]. Briefly, ~$10^7$ tachyzoites (RH strain) growing intracellularly in HFF cells in a T25 flask were incubated at 37˚C for 4 h in 0.1% Fetal Bovine Serum (FBS) DMEM growth medium containing either 2.5 mM ethyl methane sulphonate (EMS) or the appropriate vehicle controls. After exposure to the mutagen, parasites were washed three times with 1XPBS. The mutagenized population was allowed to recover in a fresh T25 flask containing an HFF monolayer without the drug for 3–5 days. Released tachyzoites were then inoculated into fresh cell monolayers in a medium containing 100 nM L35 and incubated until viable extracellular tachyzoites emerged 8–10 days later. Surviving parasites were passaged once more under continued L35 treatment and cloned by limiting dilution. Four cloned mutants were isolated, each from 6 independent mutagenesis experiments. Thus, each flask contained unique SNV pools.

## RNA-seq, sequence alignment and variant calling

For each biological assay, a T175 flask containing a confluent monolayer of HFF was infected with RH wild-type or L35-resistant strains. Total RNA was extracted and purified using TRIzol (Invitrogen, Carlsbad, CA, USA) and RNeasy Plus Mini Kit (Qiagen). RNA quantity and quality were measured using NanoDrop 2000 (Thermo Scientific). RNA sequencing was performed as previously described [37], following standard Illumina protocols, by GENEWIZ (South Plainfield, NJ, USA). Briefly, RNA quantity and integrity were determined using the Qubit Fluorometer and the Fragment Analyzer system with the PROSize 3.0 software (Agilent Technologies, Palo Alto, California, USA). The RQN ranged from 8.6 to 10 for all samples, which was considered sufficient. Illumina TruSEQ Stranded RNA library prep and sequencing reagents were used following the manufacturer's recommendations using polyA-selected transcripts (Illumina, San Diego, CA, USA). The samples were sequenced on the Illumina NovaSeq platform (2 x 150 bp, single index) and generated ~20 million paired-end reads for each sample (**S1 Table**). The quality of the raw sequencing reads was assessed using FastQC [38] and MultiQC [39]. The RNA-Seq reads (FASTQ) were processed and analyzed using the Lasergene Genomics Suite version 15 (DNASTAR, Madison, WI, USA) using default parameters. The paired-end reads were uploaded onto the SeqMan NGen (version 15, DNASTAR. Madison, WI, USA) platform for reference-based assembly and variant calling using the *Toxoplasma* Type I GT1 strain (ToxoDB-46, GT1 genome) as a reference template. The ArrayStar module (version 15, DNASTAR. Madison, WI, USA) was used for normalization, variant detection, and statistical analysis of uniquely mapped paired-end reads using the default parameters. The expression data quantification and normalization were calculated using the RPKM (Reads Per Kilobase of transcript per Million mapped reads) normalization method. Variant calls were filtered to select variants in coding regions with the following criteria: SNP% $\geq$ 90%, variant depth $\geq$ 10, and absence in the parental wild-type strain (**S1 Data**). SNVs, insertions, and deletions in regulatory or intergenic regions were filtered out as they are unlikely to contribute to drug resistance. Mutations were plotted on a Circos plot using Circa (OMGenomics.com). The Illumina RNA-seq dataset generated during this study is available at National Center for Biotechnology Information Gene Expression Omnibus (NCBI GEO): GSE217090 (https://www.ncbi.nlm.nih.gov/geo/query/acc.cgi?acc=GSE217090).

## Protein purification

PRSs from *Tg*PRS: length 334–830 (TGME49_219850 S8G8I1_TOXGM) and *Hs*PRS: length 1015–1506 (EPRS (EARS, PARS, PIG32) Q3KQZ8_HUMAN) were purchased as a gBlock (Integrated DNA Technologies, Leuven, Belgium). The ORF of the *Tg*PRS and *Hs*PRS was optimized for expression in the *E. coli* BL-21 strain. The transformed *E. coli* BL-21 strain was grown in an LB medium containing 50 μg ml$^{-1}$ kanamycin to an $A_{600}$ of 0.6–0.8 at 37˚C. Expression of both His6-tagged recombinant proteins was induced by adding 0.65 mM isopropyl β-D-thiogalactopyranoside (IPTG) to cells grown at 37˚C for 6 h, followed by incubation at 18˚C for 16–18 hr. Bacterial cells were lysed by sonication in buffer containing 50 mM Tris–HCl pH 8.0, 200 mM NaCl, 3 mM ßME, 15% v/v glycerol, 0.1 mg ml$^{-1}$ lysozyme, and EDTA free protease inhibitor cocktail (Roche). The lysed cells were cleared by centrifugation at 20,000 g for 45 min, and the supernatant was affinity captured using nickel-nitrilotriacetic acid-agarose beads (Ni-NTA) (GE Healthcare), followed by elution with buffer containing 50 mM Tris–HCl pH 8.0, 200 mM NaCl, 10 mM ßME and 250 mM Imidazole. The eluted protein fractions were dialyzed against 30 mM HEPES pH 7.5, 20 mM NaCl, 1 mM DTT and 0.5 mM EDTA (buffer A). The protein was purified by heparin chromatography (GE Healthcare) using NaCl gradients with buffer B containing 30 mM HEPES pH 7.5, 500 mM NaCl, 1 mM DTT, and 0.5 mM EDTA. The protein peak was found at 40% buffer B. The 6xHis tag was removed by incubating with TEV protease at 20˚C for 24 h. The cleaved *Tg*PRS and *Hs*PRS proteins were concentrated with a 30-kDa cut-off Centricon centrifugal device (Millipore) followed by gel permeation chromatography on a Superdex 200 column 16/60 column (GE Healthcare) in a buffer containing 20 mM HEPES pH 7.5, 200 mM NaCl, and 2 mM DTT. Bovine serum albumin (66 kDa, Sigma) was used as a standard for molecular mass estimation. The eluted fractions were checked by SDS-PAGE, and the pure fractions were pooled, concentrated in 50 mM Tris–HCl pH 8.0, 200 mM NaCl, 10 mM ßMe and stored at -80˚C.

## Thermal shift assays

Fluorescence-based thermal shift assays were performed to assess the binding potencies of the five 1-(pyridin-4-yl) pyrrolidin-2-one (PPL)-based derivatives for *Tg*PRS and *Hs*PRS in the presence or absence of substrates (L-Pro and ATP). Purified PRS enzymes in the presence and absence of their substrates and inhibitors were heated from 25 to 99˚C at a rate of 1˚C min$^{-1}$, and fluorescence signals of the SYPRO orange dye were monitored by a quantitative real-time PCR system (Life Technologies). Proteins were used at 1 μM, the drugs at 50 μM (for *Tg*PRS) and 100 μM (for *Hs*PRS), and the substrates at saturating concentrations of 2 mM. The melting temperature is an average of three measurements, and data were analyzed using Protein Thermal shift software (v1.3, Thermofisher). The inhibitors and substrates alone in assay buffers and no PRS enzyme controls were used, and flat lines were observed for these fluorescence readings across the temperatures. The derivative $T_m$ was used for analysis.

## Enzyme inhibition assays

Aminoacylation inhibition assays were performed according to the previously published report [7,40]. Briefly, 25 μM ATP, 25 μM L-pro, and 400 nM recombinant purified protein (*Tg*PRS and *Hs*PRS) in a buffer containing 30 mM HEPES (pH 7.5), 150 mM NaCl, 30 mM KCl, 50 mM, MgCl$_2$, 1 mM DTT and 2 U/ml *E. coli* inorganic pyrophosphatase (NEB) at 37˚C was used to perform the assay. The assay was carried out in a transparent, flat-bottomed, 96-well plate (Costar 96-well standard microplates) for 1.6 h at 37˚ C. Malachite green solution was added to stop the enzymatic reaction. Absorbance was measured at 620 nm using a Spectramax M2 (Molecular Devices). The background controls were set up without the *Tg*PRS and

*Hs*PRS enzymes. The values obtained were deducted from the enzymatic reaction values. The five compounds were added to the aminoacylation assay reaction buffer in concentrations of 0.00005 to 50 μM. The $IC_{50}$ values for the data are shown for three replicates as the mean ± SD. All competitive assays were performed in the presence of both L-pro and ATP at 1.5X concentrations of their $K_m$ values.

## Crystallization

Highly purified *Hs*PRS (12–15 mg mL$^{-1}$) and *Tg*PRS (10–14 mg mL$^{-1}$) enzymes were used for crystallization via the hanging-drop vapor-diffusion method at 20˚C using commercially available crystallization screens (Hampton Research and Molecular Dimensions). Compounds were initially in 100 mM stocks prepared in 100% DMSO–which were then added to the protein whilst crystallization with the final DMSO concentration being 2.5–7.5%. Initial screening was performed in 96-well plates using a nanodrop dispensing Mosquito robot (TTP Labtech). Three different drop ratios of purified protein and reservoir (i.e., 1:1, 2:1, and 1:2 drop ratios) were used for the crystallization trials. Each drop was equilibrated against 100 μl of the corresponding reservoir solution. Before crystallization, 1 to 3 mM compounds and 2 mM L-pro were added to PRS enzymes, and the mixtures were incubated at 4˚C for 10 min. Diffraction-quality crystals were obtained at 20˚C by the hanging-drop vapor diffusion method. The crystallization conditions for each enzyme-inhibitor complex are listed in **S2 Table**.

## Diffraction data collection and structure determination

The X-ray diffraction data sets were collected on beamlines at Diamond Light Source (DLS), United Kingdom, PROXIMA 1 and PROXIMA 2A, SOLEIL, France. The data were processed by the auto-processing pipelines using DIALS and XDS for integration [41]. The initial models were determined by the molecular-replacement (MR) method using Phaser [42] and the available host (PDB ID: 4K86) and parasite PRS (PDB ID: 5XIF) enzyme structures as the template. The structures were further refined by iterative refinement cycles with Phenix [43] and model building with COOT [44]. Map interpretations and model building were based on electron densities in difference Fourier ($F_o - F_c$) and $2F_o - F_c$ maps. In all stages, model building was guided by manual inspection of the model and $R_{free}$. The substrate/inhibitor and water molecules were added based on the difference Fourier maps ($F_o - F_c$). The occupancies of the ligand molecules were refined, and highly disordered loop regions were not included in the final model. The stereo-chemical quality of the models was assessed and corrected using MolProbity [45]. The summary of the collection and refined parameters are given in **S3 and S4 Tables**. The figures were prepared using Chimera [46] and PyMOL [47]. Structural interaction analyses were performed using the PLIP server [48].

## Accession numbers

The atomic coordinates and structure factors for the *Tg*PRS and *Hs*PRS complex with inhibitor-AMPNP/L-pro are deposited in the RCSB Protein Data Bank (PDB), and the accession codes are listed in **S2 Table**.

## Results

### T. gondii growth inhibition assays

1-(Pyridin-4-yl) pyrrolidin-2-one (PPL) derivatives are ATP mimetics that were designed as anti-fibrosis scaffolds that selectively target *Plasmodium* prolyl-tRNA synthetase (PRS) [29,31,49]. Efficient *in vitro* inhibition of *T. gondii* growth was repeatedly confirmed by

probing the half-maximum effective concentrations for all compounds tested here ($EC_{50}$) [L95: 277 ± 70 nM, L96: 106 ± 35 nM, L97: 544 ± 130 nM, L35: 27 ± 7 nM, L36: 5800 ± 350 nM] (**Figs 2A and S1A**). Among these five PPL derivatives, the $EC_{50}$ of L35 with 27 nM was ~14X lower than pyrimethamine with an $EC_{50}$ of 396 nM—the standard care for toxoplasmosis. Therefore, we further investigated L35 to assess its efficacy. The growth of the type I RH strain was monitored using human foreskin fibroblasts (HFFs) treated with L35; pyrimethamine and vehicle (DMSO) were used as positive and negative controls respectively. Complete and sustained growth inhibition was observed at 100 nM of L35 with no adverse effects on host cells (**Fig 2B**). The selectivity index (SI) of the five PPL derivatives was determined using ARPE19 (epithelial cells) and MDA321 (breast cancer) cell lines (**Figs 2C, 2D, S1D and S1E**). Cells were incubated with increasing concentrations of drugs for 72 h. The viability of the human cell lines was determined to calculate the cells' cytotoxicity concentration, $CC_{50}$. SI values presented in the tables are based on the $CC_{50}$ of the human cells divided by the $EC_{50}$ of *T. gondii*. The PPL derivative—L36, with the highest $EC_{50}$ value of 5800 nM, showed no inhibition for the human cell lines at 1000 nM (**Figs 2A and S1A**).

## Target validation of the PPL derivatives via random T. gondii mutagenesis

We then investigated the mechanism of action of L35 as it displayed the lowest $EC_{50}$ at 27 nM. This was done via a forward genetic screen using chemical (ethyl methane sulphonate (EMS)) mutagenesis to isolate L35-resistant parasites. Resistant parasites were analyzed with the wild-type strain by RNA-seq to identify mutations in the coding sequence conferring drug resistance by computational analysis of the variants (**Fig 2E**). To map the EMS-induced mutations conferring L35 resistance, Illumina sequencing reads were aligned to the ~65-Mb *T. gondii* GT1 reference genome. The assembled sequences were analyzed to identify single nucleotide variations (SNVs), small insertions, or short deletions using the parental strain as a reference, as described previously (**Fig 2F**) [37]. By focusing on mutations in coding sequences, we identified a single gene, *PRS*, that contained two SNVs, resulting in amino acid substitutions at the protein residues numbered 477 and 592 (**S1 Table**). The mutations T477A and T592S in the L35-resistant lines were absent in the parental strain. To confirm that the *PRS* mutations were sufficient to confer resistance to L35, we reconstructed each of the mutations identified in L35-resistant parasites into the susceptible parental wild-type strain using the CRISPR/Cas9 system coupled with homology-directed repair for gene editing in *T. gondii* (**S2A Fig**). After selection with L35, emerging resistant parasites were cloned, and DNA sequencing established that the mutations were correctly inserted into *TgPRS* (**Fig 3A**). In the genetically modified parasites, the PRS mutations T477A and T529S substantially reduced susceptibility to L35 compared to wild-type parasites (**Figs 3B–3D and S2B and S5 Table**). Notably, the plaque size of the T592S mutant was somewhat higher than that of the T477A mutant, however, upon treatment with L35, the trend was reversed (although the differences were non-significant). Furthermore, we would like to point out that the mutant T477A was ~ 3X more resistant towards L35 (186 nM for T477A vs 76 nM for T592S) than the T592S mutant. However, both these mutants (T477A and T592S) are equivalently susceptible to pyrimethamine as wild-type parasites (as is expected by virtue of the mutual exclusion of their mechanisms of action). The activity of the other four compounds was tested against the PRS-edited parasites (**Figs 4, S1B, and S1C and S5 Table**). Although there are some differences between the $EC_{50}$ values for the five compounds in the L35-resistant parasites—T477A and T592S, the $EC_{50}$ values were higher than in the wild-type parasites confirming that these five PPL-derivatives are active against PRS (**Fig 4A–4E and S5 Table**). Moreover, *Tg*PRS mutation T477A substantially reduced the sensitivity to L36 in the engineered compared to wild-type parasites (**Fig 4D and S5 Table**).

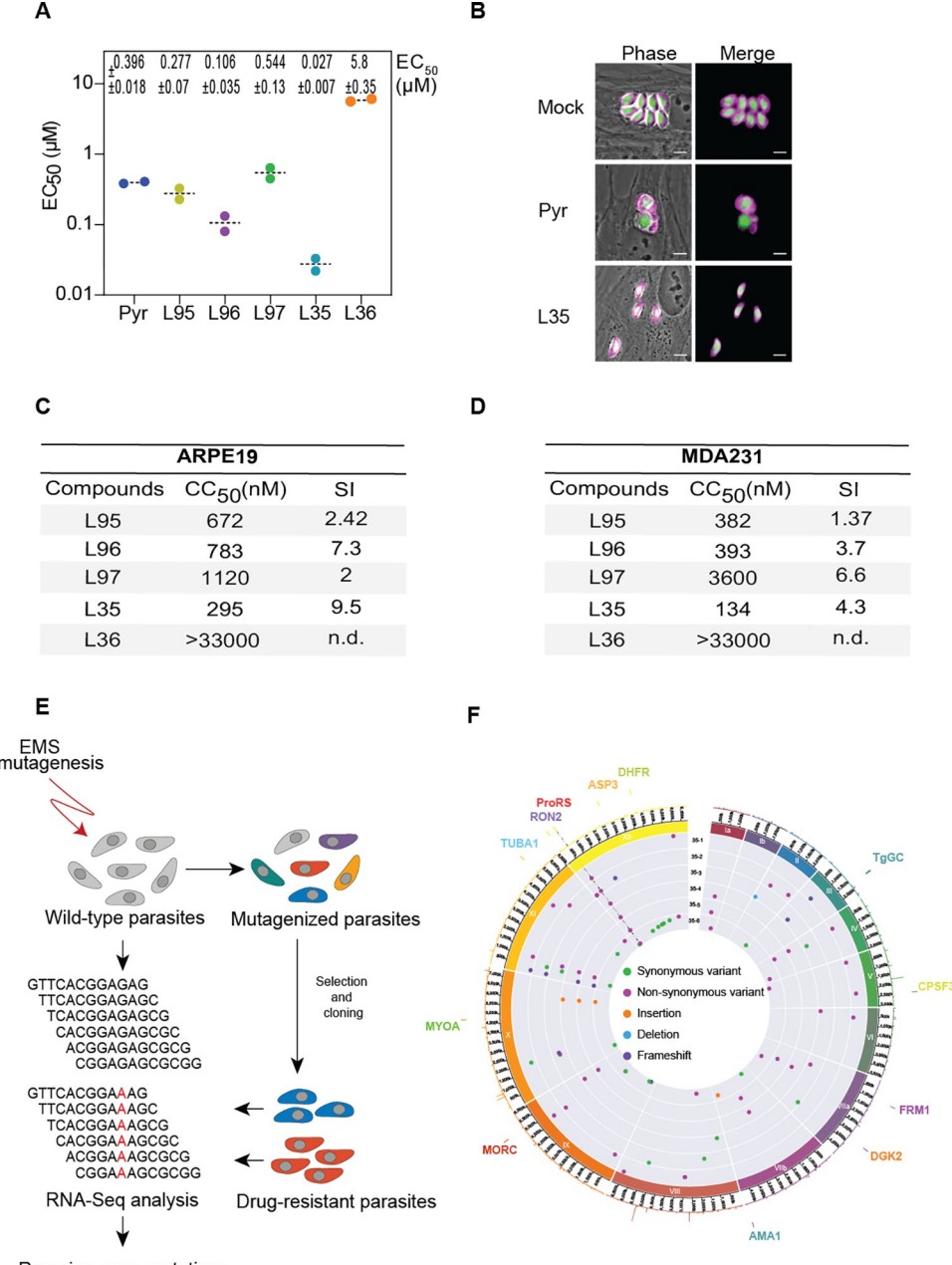

**Fig 2. Efficiency of the five PPL-derivatives on *Toxoplasma gondii* parasite. A)** $EC_{50}$ values for the indicated compounds were determined *in vitro* using *T. gondii* tachyzoites (RH Δ*ku80 UPRT::NLuc-P2A-EmGFP*). Data are mean±SD of at least 2 independent biological replicates, each with 3 technical replicates. The corresponding dose-response curves are shown in S1A Fig **B)** IFA showing the efficiency of the L35 compound. HFFs were infected with tachyzoites (RH Δ*ku80 UPRT::NLuc-P2A-EmGFP*) and incubated with 1 μM of pyrimethamine, 0.1 μM of L35 or 0.1% DMSO as control. Cells were fixed 24 h post-infection and then stained with antibodies against the *T. gondii* inner membrane complex protein GAP45 (magenta). The cytosolic GFP is shown in green. Scale bars, 5 μm. **C-D)** The Selectivity Index (SI) of the five PPL-derivatives was determined using the cell lines (C) ARPE19 (epithelial cells) and (D) MDA321 (breast cancer). Cells were incubated with increasing concentrations of drugs for 72 h. The viability of the human cell lines was determined to calculate the cytotoxicity concentration of the cells, $CC_{50}$. SI values, presented in the tables, are based on the human cells $CC_{50}$ divided by the *T. gondii* $EC_{50}$. No inhibition was detected for L36 compound. n.d., not detected. Associated dose-response curves are showed in S1 Fig. **E)** The diagram shows the strategy for L35 target deconvolution. The workflow is based on forward genetic screen using chemically induced resistant mutant lines. EMS-mutagenized parasites were then selected with lethal concentrations of L35. The resistant parasites along with the wild-type strain were analyzed by RNA-seq for computational analysis of the variants to

identify the mutations on coding sequence conferring drug-resistance. **F)** Circos plot showing the single nucleotide variants (SNVs), insertions, and deletions detected by transcriptomic analysis of the *T. gondii* L35-resistant lines, grouped by chromosome (numbered in Roman numerals with size intervals given outside). Each dot in the 6 innermost gray tracks corresponds to a scatter plot of the mutations identified in the coding regions of the 6 drug-resistant strains, with each ring representing one of the 6 drug-resistant lines (35–1 to 35–6). In the second outermost track, lines depicting whole-genome RNA-seq data of the *T. gondii* parental strain (RPKM values of genes are shown). Each bar in the outermost track represents locations of selected essential genes. (n.d–not detected).

Our group recently reported the simultaneous binding of L95 and HFG in complex with *Tg*PRS covering all three of the enzyme-substrate subsites, with HFG occupying the 3'-end of tRNA and the L-pro binding sites, and L95 resident in the ATP pocket (**Fig 4F**). Structural data analysis for the available three-dimensional crystal structure of the complex *Tg*PRS-L95 +HFG (PDB: 7EVU) revealed that the two L35-resistant mutations–T477A and T592S –are proximal to the two ends of the scaffold (**Fig 4F**). Interestingly, the mutation T592S made the parasite considerably more susceptible towards L36 (See **S5 Table**). Although the variant residues *Tg*- T477 and T592 were not directly in the ATP, tRNA, or L-pro site, these were located at the edge of the ATP pocket. This indicates that both the methyl-pyrazole and hydroxy-methyl terminals may play critical roles in the selectivity of the PPL scaffolds. Based on these observations, we biochemically and structurally characterized each of these five PPL derivatives in *Tg*PRS and *Hs*PRS enzymes via enzyme-based aminoacylation inhibition assays and high-resolution crystal structures of drug-bound enzymes, respectively.

## The five PPL-derivatives inhibit aminoacylation via competition with ATP

The potency of the five PPL derivatives against *Tg*PRS and *Hs*PRS was evaluated using enzyme-inhibitor binding and inhibition assays. We examined the first step of the aminoacylation reaction, i.e., the formation of the aminoacyl-adenylate complex, and measured the release of pyrophosphate (PPi) using a malachite green dye-based assay. The measured half-maximal inhibitory concentration ($IC_{50}$) values for each inhibitor L95, L96, L97, L35, and L36 are shown in **Fig 5**. The high potency of these inhibitors was evident from their nanomolar $IC_{50}$ values for *Tg*PRS, i.e., L95: 139.9 nM, L96: 79.7 nM, L97: 50 nM, L35: 9.2 nM, and the exception of L36 with $IC_{50}$ of 2409 nM (**Fig 5A and 5B**). Interestingly, the ATP-mimetic PPL scaffolds were originally designed as antifibrosis scaffolds against *Hs*PRS. However, three out of five PPL derivatives, i.e., L97, L96, and L35, were found to be selective against *Tg*PRS as opposed to its human counterpart *in vitro*, with enzymatic selectivity indices of 27.9, 13.5, and 8.2, respectively (**Fig 5C**).

Enzyme-ligand interactions are critical for stabilizing the enzyme conformation during the thermal denaturation process and increasing the melting temperature (Tm). Our analysis of the thermal shift data for enzyme-inhibitor binding was consistent with the inhibition data. It showed that L97 and L35 were bound more efficiently to *Tg*PRS in the presence of L-Pro compared with *Hs*PRS (**Fig 5D**). For *Tg*PRS, we observed ΔTm (˚C) values of 22, 23, 20, 26, and 14 for L95, L96, L97, L35, and L36, in the presence of L-Pro, respectively, compared to the apo-*Tg*PRS. Similarly, the ΔTm (˚C) values for *Hs*PRS were 19, 21, 16, 23, and 10 for L95, L96, L97, L35, and L36, in presence of L-pro, respectively, compared to the apo-*Hs*PRS (**Fig 5D**). Notably, the five PPL-derivatives stabilized the PRSs more effectively in the presence of L-Pro, with L35 being the most stable (ΔTm of 26 and 23 (˚C) for *Tg*- and *Hs*-PRSs), suggesting a cooperative binding that assists the five inhibitors in binding the enzyme more efficiently in the presence of the L-amino acid, a natural substrate (**Fig 5E**). Next, co-crystallization was performed to elucidate the structural basis of activity of the five PPL derivatives and PRS interactions.

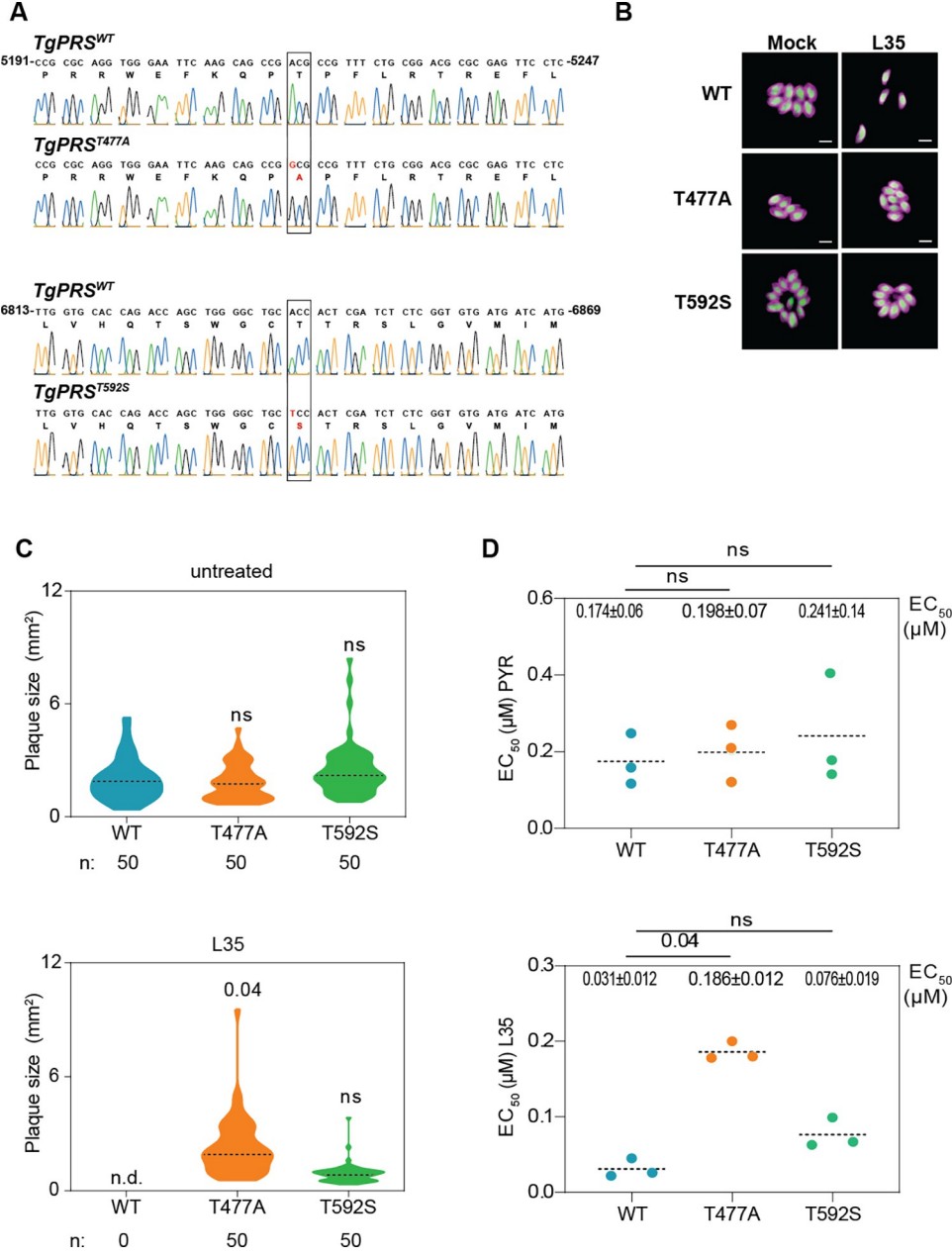

**Fig 3. Validation of PRS molecular target. A)** Sanger sequencing chromatogram showing *PRS* editing. Nucleotide positions relative to the ATG start codon on genomic DNA are indicated. A schematic of the strategy used for the CRISPR-mediated gene editing is represented in S2 Fig. **B)** Fluorescence microscopy showing intracellular growth of WT and the *PRS*-edited parasites (T477A and T592S). Confluent HFFs were infected with parental or recombinant RH tachyzoites (RH Δ*ku80 UPRT::NLuc- P2A-EmGFP*) and incubated with 0.1 μM of L35 or 0.1% DMSO as control. The cells were fixed 24h post-infection and stained with anti-GAP45 antibody (magenta). Cytosolic GFP is shown in green. Scale bars, 5 μm. **C)** Effects of L35 and PRS mutations on *T. gondii* lytic cycle were determined by plaque assay. Plaques sizes were measured after 7 days of growth in the presence of the drug or 0.1% DMSO. The graphs represent the size of 50 plaques when detected, and n.d., when not detected. Associated data are in S2 Fig. **D)** $IC_{50}$ values for Pyrimethamine and L35 were determined for wild-type (WT) and the engineered *PRS* mutant strains (T477A and T592S). The mean ±SD of two independent experiments is shown. On the top of each graph the significant difference is indicated. Statistical analyses were performed by one-way ANOVA test using GraphPad8 software. n.s., not significant.

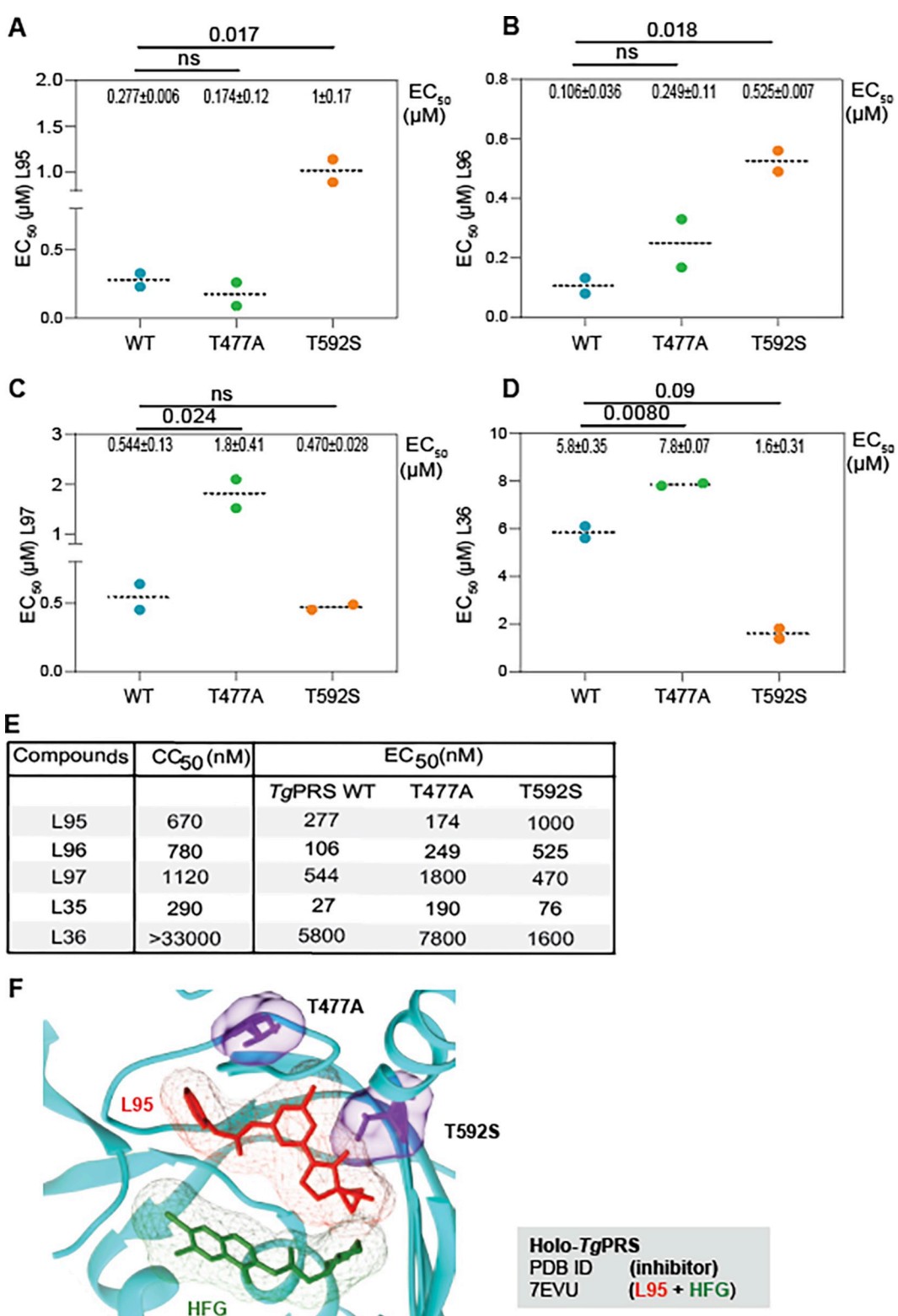

**Fig 4. Activity of five compounds on *PRS* edited parasites. A-D)** EC$_{50}$ values for L95, L96, L97, and L36 were determined for wild-type (WT) and the engineered *PRS* mutant strains (T477A and T592S). Data are mean value ±SD from at least 2 independent biological replicates, each with 3 technical replicates. Associated dose-response curves are shown in S2C Fig. Higher EC$_{50}$ values for the mutants confirm that PRS is the molecular target of all five compounds tested. The *P-values* were calculated using one-way ANOVA analysis and multiple-comparison *post hoc* tests. n.s., not significant. Associated dose-

response curves are shown in S1 Fig. **E)** The $EC_{50}$ (nM) values for the engineered *PRS* mutant strains (T477A and T592S) are shown in comparison with the *Tg*PRS wild type. **F)** The catalytic cavity of *Tg*PRS is shown with two inhibitors along with their electron densities–HFG (green) and L95 (red)–bound to the tRNA and L-pro binding sites and the ATP binding site respectively. The ribbon of PRS is in cyan whilst the resistance-conferring mutating residues T477 and T592 to L35 are shown in purple. (n.d.–not detected).

## Structural overview of PRS with the five PPL-derivatives

To decipher the structural basis of the selectivity of the inhibitors by the host (*Hs*) and parasite (*Tg*) PRSs, and to address the apparent higher affinity of the drug in the presence of L-pro, we attempted to co-crystallize the *Tg*- and *Hs*-PRSs with the five PPL derivatives. We successfully co-crystallized all five compounds with *Tg*PRS (L95, L96, L97, L35, and L36) and three with *Hs*PRS (L95, L96, and L97), all in complex with L-pro (S2 Table). Interestingly, co-crystals of *Hs*PRS with these compounds were obtained in two forms, namely forms 1 and 2, in 0.1 M HEPES pH 7.5, 20% PEG 3350 and 0.5 or 1.5 M $CaCl_2$ (S2 Table). Form 1 crystals belonged to the monoclinic space group $P2_1$ (a = 71, b = 91, c = 83 Å and β = 110˚) with two molecules per asymmetric unit, and form 2 crystals belonged to the orthorhombic space group $P2_12_12_1$ (a = 70, b = 106, c = 145 Å), except for *Hs*PRS-L97. The RMSD after superposition of Cα-atoms of the two molecules of monoclinic form is 0.3–0.5 Å. The RMSD after superposition of the orthorhombic forms is 0.2–0.4 Å for $C_\alpha$-atoms. The RMSD between the molecules of form 1 and form 2 is 0.5 Å. The RMSD between the ligand-bound molecules in monoclinic or orthorhombic forms is ~0.2–0.5 Å. The RMSD between the empty and L-pro bound *Hs*PRS molecules with ligand-bound molecules are in the range of 0.5–0.7 Å. The B-factor values (S3 Table) revealed that most structures are well ordered and have low mobility. The RMSD between the empty and L-pro bound *Tg*PRS and with ligand-bound *Tg*PRS is 0.6–0.8 Å and 0.2–0.5 Å, respectively. The overall folding of the host (*Hs*) and parasitic (*Tg*) PRSs was similar to the previously reported structures (PDB ID: 4K86, 5XIF). Minimal but significant conformational changes were observed for the active site residues and a shift in the active site loops. These are described and discussed in the following sections. The bound inhibitor molecules were verified using a difference Fourier electron density map at 3 σ levels (Fig 6). Because the RMSD between the present enzyme-inhibitor complex structures with known structures was <1 Å, we considered only chain A for all structural descriptions.

## Toxoplasma gondii PRS (TgPRS)

The three-dimensional crystal structures of all the five PPL derivatives in the holo-*Tg*PRSs were probed (Fig 7A). The five PPL derivatives were buried in a ligand-induced fit model between the ATP pocket (470–483) and the loop (528–538) (Fig 7B). Superposition of apo-(PDB: 5XIF) (in grey) and the holo-*Tg*PRSs (in purple) and structural analysis revealed remarkable plasticity in the ATP pocket (470–483) of *Tg*PRS to accommodate these PPL-derivatives in a groove forming the binding pocket that is composed of Arg470, Glu472, Lys474, Arg481, Thr482 (Fig 7C). The second half of the binding pocket consists of Phe485, Phe 534, Gln555, Thr592, and Arg594 (Fig 7D and 7E). The phenyl ring of Phe485 and the guanidinium moiety of Arg594 support the adenine equivalent of the 6-methylpyridine core via a π-π stacking interaction. The compounds are stabilized by hydrogen bonds via the O atom of the phenylacetamide group and the keto O atom of oxopyrrolidine interaction with a water molecule. Furthermore, in L35 and L95, the 6-methylpyridine N atom interacts with the main-chain N atom of Thr482, Benzyl-formamide (L96, L97, L35, L36) or phenylacetamide (L95) N-atom interacts with the OG1 of side-chain Thr482. The oxopyrrolidine keto O atom interacts with the OG1 of side-chain Thr592 in L95, L96, and L97. Additionally, the O atom of

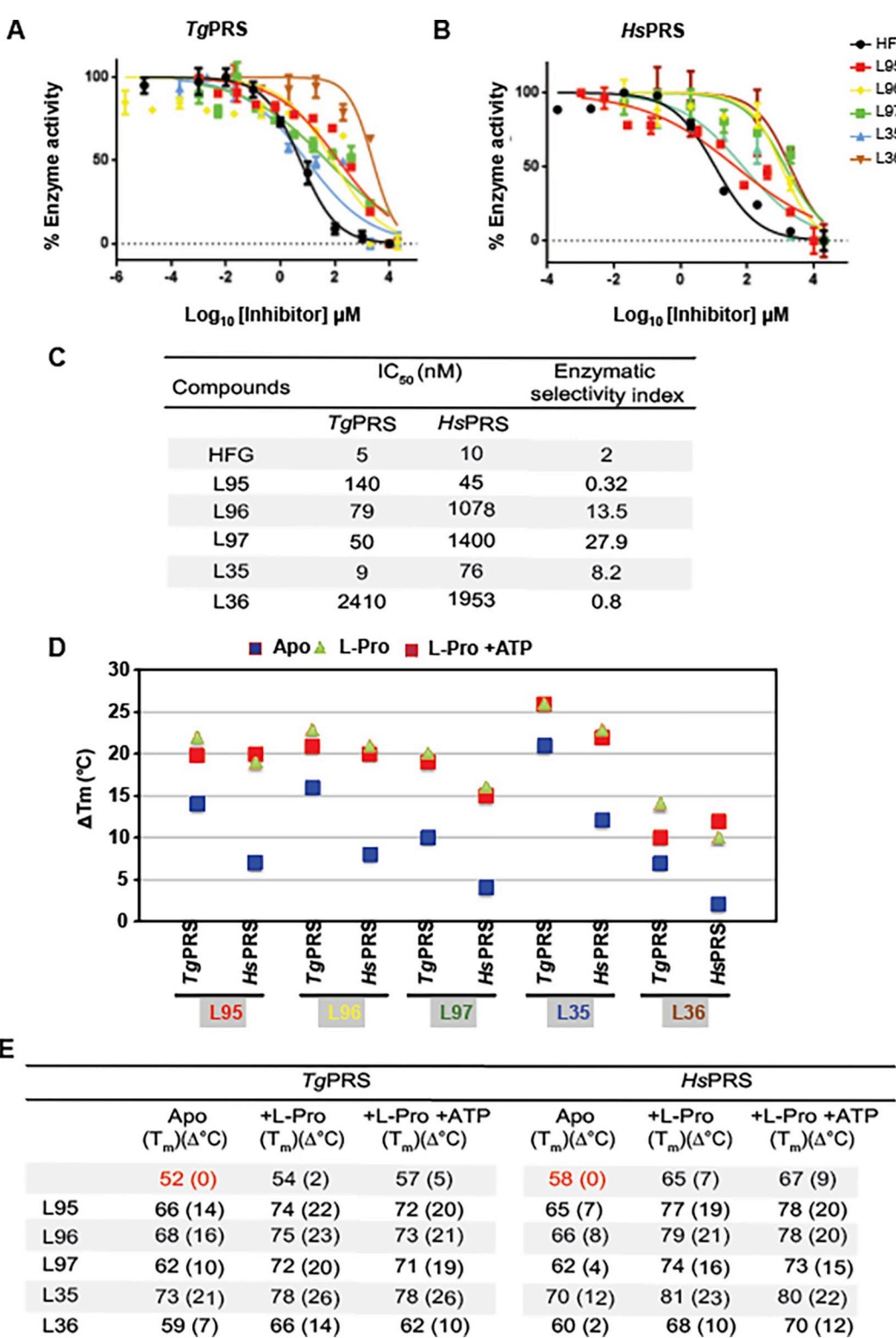

**Fig 5. In vitro assessment and binding affinity.** Aminoacylation activity inhibition assays were performed for (**A**) $Tg$PRS, and (**B**) $Hs$PRS with HFG and the five compounds. The dose-dependent sigmoidal curves were used to calculate $IC_{50}$ values for each compound for the respective PRSs. Error bars represent mean ± SD. Data are shown as mean ± SD (n = 3 independent experiments). (**C**) The $IC_{50}$ values (nM) are shown. (**D/E**) Thermal shift assays were performed to evaluate the melting temperature-based binding affinity for each of the five compounds with $Tg$PRS and $Hs$PRS. The five compounds

showed similar trends based on the melting temperature shifts. Thermal shift profiles for the *Tg*PRS and *Hs*PRS enzymes with the five PPL derivatives are shown.

Hydroxymethyl attached to the pyrrolidine group interacts with OG1 of Thr558 and the side chain of Gln555 in L97 (**S3 Fig**).

The superposition of holo-*Tg*PRS revealed no striking steric or sidechain rotameric conformational differences among the five PPL derivatives bound structures of *Tg*PRS complexes. In contrast to L35, L95, L96, and L97, compound L36 showed poor enzyme inhibition as indicated by an IC50 value in the μM range compared to the other four compounds analyzed in the current study (**Fig 5A and 5C**). This is attributed to preferential binding favoring the (S) enantiomer (L35) and not the (R) enantiomer (L36). Structurally, the stacking of 4-(3-Fluorophenyl)-1-methylpyrazole moiety allows rotameric conformational changes in Glu472,

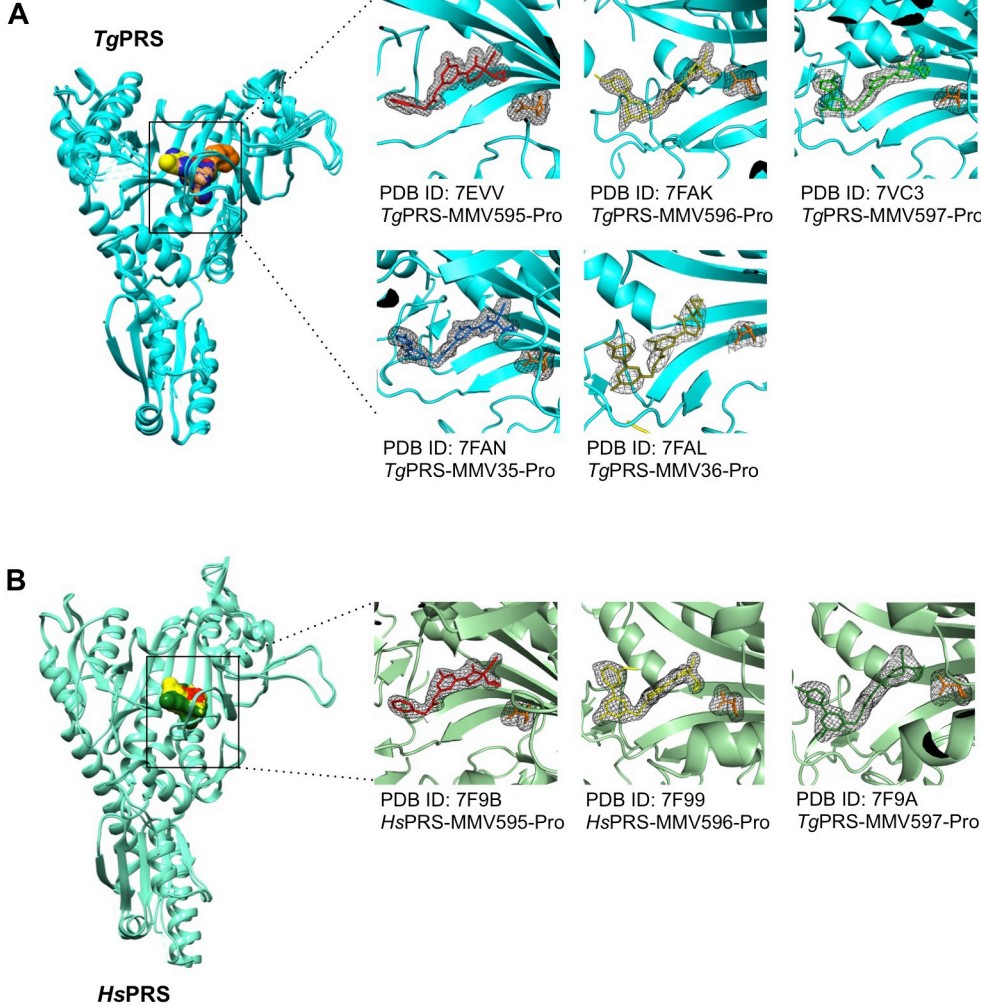

**Fig 6. Crystal structure of PRSs in complex with the five compounds.** The X-ray structures of the **(A)** *Tg*PRSs (cyan) in complex with the five compounds (L95-red; L96-yellow, L97-green, L35-blue and L36-brown). **(B)** *Hs*PRS (in palegreen) in complex with the three compounds (L95-red; L96-yellow, L97-green). A composite OMIT difference Fourier map (F$_o$-F$_c$) generated at respective resolution (Å) and contoured at 3 σ showing ATP-mimetics-bound *Tg*PRS and *Hs*PRS are shown.

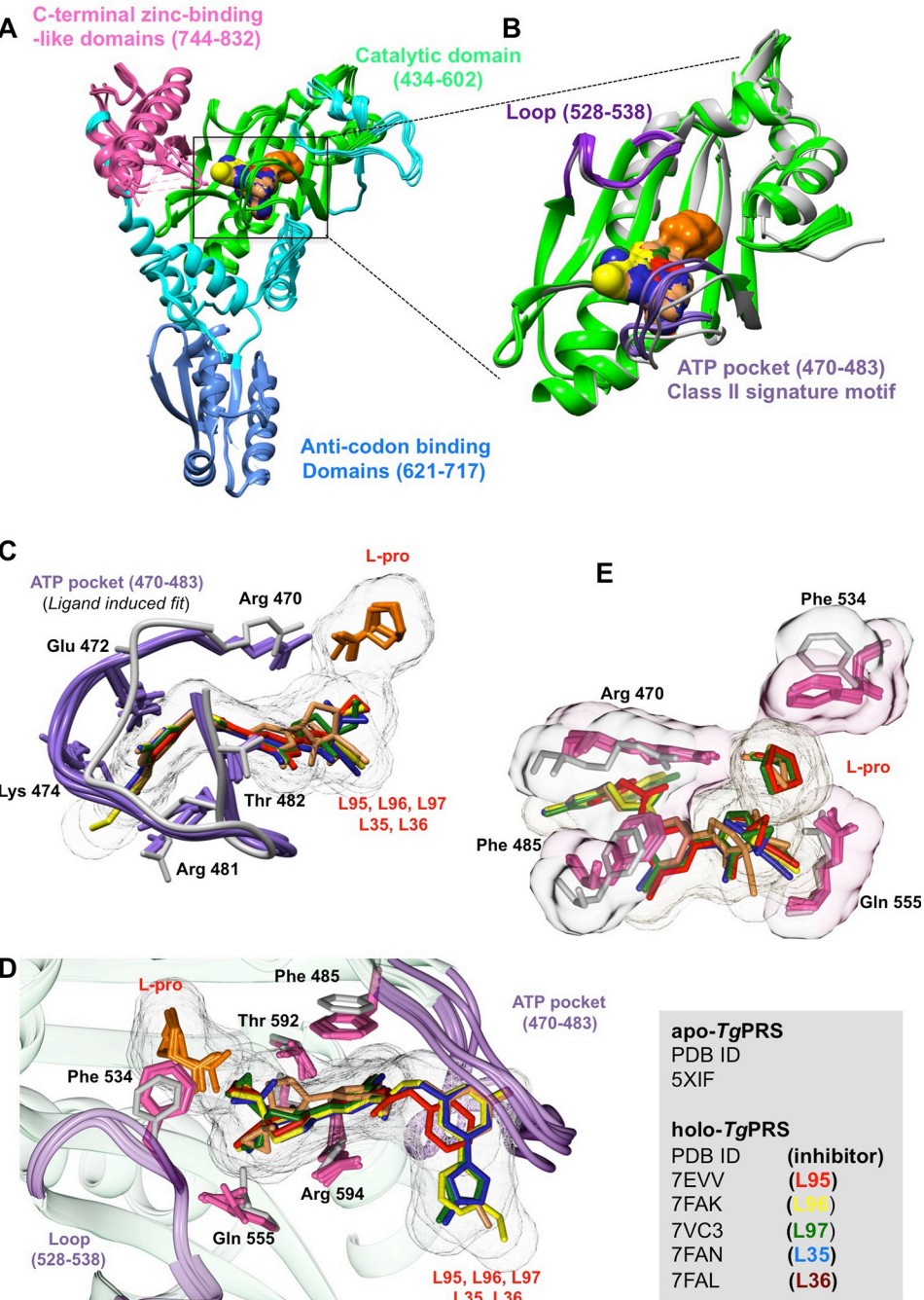

**Fig 7. The five inhibitors in complex with *Tg*PRS.** Overall structures of L95-, L96-, L97-, L35-, and L36-bound *Tg*PRSs. **(A)** *Tg*PRS-holo can be divided into three domains. The catalytic domain (434–602), anti-codon binding domain (621–717) and C-terminal zinc-binding-like domain (744–832). **(B)** The ATP pocket (470–483) class II signature motif and the loop (528–538) are shown in purple. **(C)** Structural comparison of the *Tg*PRS-apo (in grey) and the ATP-mimetics-bound *Tg*PRS-holo (in purple) is shown. **(D and E)** The residues forming the cavity compound and the L-pro cavity are shown.

Lys474, and Arg481, wherein Arg481 is stabilized by the H-bonds in L95/L96/L97/L35 complexes (**S3A–S3D Fig**) but not in L36 (**S3E Fig**). The stereoisomeric configuration of cyano- and cyclopropyl groups in L36, in contrast to L35 (**Fig 1**), creates torsion at ~90˚ in the oxo-

pyrrolidine and methyl-pyridine groups. Regardless, the 6-methylpyridine core is stacked between the phenyl ring of Phe485 and the guanidinium moiety of Arg594. Whereas the significant ~90˚ bent in oxo-pyrrolidine exposes its hydrophobic core to the active site, displacing the oxo-pyrrolidine keto O atom away from Thr592 and surrounding water molecules. Interestingly, the stereoisomeric distortions in L36 result in a slight lateral shift in the neighboring L-pro, preventing its OXT atom from forming a fourth H-bond with NE2 of His560. This absence of the fourth H-bond may lead to loose packing in the active site.

## Homo sapiens PRS (HsPRS)

The complex structure of *Hs*PRS with L95, L96, and L97 was determined to address the structural basis for differential inhibition (**Figs 6B and 8A**). Significant structural variations within the catalytic domain (1016–1296): ATP pocket (1152–1165), loop (1210–1220), and motif (1084–1115) were observed for the three co-crystallized structures (**Fig 8B**). The loop 1152–1165 exhibited ligand-induced fitting, and the residues Arg1152, Trp1153, Phe1155, Lys1156, and Arg1163 formed a cavity-like structure to fit L95, L96, and L97. These three compounds displayed side-chain rotameric conformational changes relative to the apo-*Hs*PRS shown in **Fig 8C**. The L-pro pocket exhibited minimal structural and rotameric changes. L-pro was mainly buried within a cavity formed by Glu1123, Phe1167, Trp1169, Glu1171, His1173, Phe1216, His1242, Thr1276, and Arg1278 (**Fig 8D**).

The active site residues between *Hs*PRS and *Tg*PRS are highly conserved with a few exceptions- His1157 (Gln475), Gln1159 (Thr477), Gly1238 (Ala556), and Gly1239 (Ala557). Based on the *Tg*PRS enzyme data, L97 showed the highest selectivity index of 27.9 compared with the other four PPL derivatives. Structurally, in both *Tg*PRS and *Hs*PRS complex, L97 shared conserved interaction points except the H-bond between the O atom of hydroxymethyl attached to the pyrrolidine group with OG1 of Thr558, which is missing in *Hs*PRS (**S3 and S4 Figs**). This is due to a slight displacement of the hydroxyl group and, in turn, a slight downward displacement of the entire cyclopropyl-oxo-pyrrolidine group, sufficient enough to overcome the hydrophobic environment resulting from the exposed cyclopropyl core. This allows the accommodation of a water molecule and the restoration of its H-bond with L-pro. In addition, the guanidinium moiety of Arg1163 is slightly displaced away from the pyrazole ring to form an H-bond with the OE1 atom of neighboring Gln1159 resulting in loose stacking of the entire 4-(3-Fluorophenyl)-1-methylpyrazole moiety in the ATP pocket. This is also evident in the rotameric conformations in Lys1156 and Phe1155. Furthermore, between the two variant residues, T477 is located at the edge of the ATP pocket, while the other variant residue T592 lies on the other end facing the oxo-pyrrolidine moiety. This observation indicates that both the methyl-pyrazole terminal and hydroxy-methyl end might play a crucial role in the selectivity of L35.

## Discussion

Aminoacyl-tRNA synthetases (aaRSs) are being explored as effective, novel, and specific targets for human parasitic diseases [2,3,10,20,50,51]. Among the aaRSs, prolyl-tRNA synthetase (PRS) is a validated drug target in *Plasmodial* spp and *Toxoplasma gondii* (*Tg*) [26–31]. In the current study, we examined five ATP-mimetics L95, L96, L97, L35, and L36, based on the 1-(pyridin-4-yl) pyrrolidin-2-one (PPL) scaffold. The potencies of the five compounds against *Tg* parasites and the *Tg*PRS enzyme were validated via cell-based and enzymatic assays, respectively. The cellular ($EC_{50}$) and enzymatic ($IC_{50}$) data revealed that these drugs are effective at nM concentrations, except for L36 (**Figs 2, 3, and 5**). With an EC50 of 27 nM and cellular SI values of 9.5 and 4.3 for ARPE19 and MDA231 cell lines, L35 was the most potent and selective

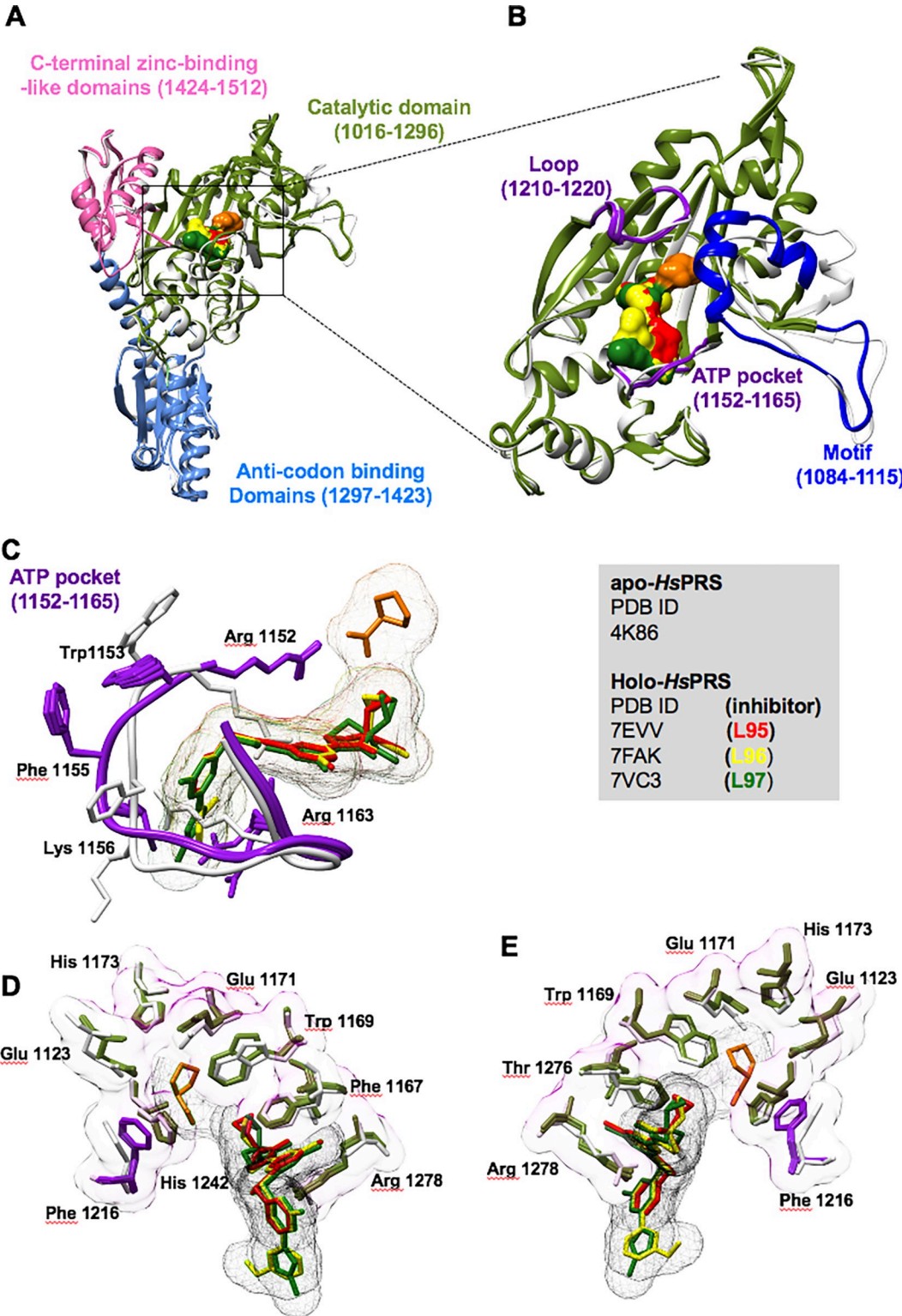

**Fig 8. The five inhibitors in complex with *Hs*PRS.** Overall structure of L95-, L96-, L97-bound *Hs*PRSs. **(A)** *Hs*PRS-holo can be divided into three domains. The catalytic domain (1016–1296), anti-codon binding domain (1297–1423), and C-terminal zinc-binding-like domain (1424–1512). **(B)** The ATP pocket (1152–1165) and the loop (1210–1220) are shown in purple. **(C)** Structural comparison of the *Hs*PRS-apo (in grey) and the ATP-mimetics-bound *Hs*PRS-holo (in purple) is shown. **(D and E)** The residues forming the cavity compound and the L-pro cavity are shown.

against *T. gondii*. In contrast, the enzymatic SI of L35 was 8.2 versus the enzymatic SI of L97 which was 27.9 (**Fig 5**). Further, the data revealed that except for L36, the other four PPL-derivatives are effective at nM concentrations against *Tg*PRS enzyme. This is attributed to preferential binding favoring the (S) enantiomer (L35) and not the (R) enantiomer (L36), as supported by the crystal structures of drug-bound enzyme complexes. To determine the mechanism of action using L35, cell-based chemical mutagenesis was employed via a forward genetic screen. *Tg*-resistant parasites were analyzed with the wild-type strain by RNA-sequencing to identify mutations in the coding sequence conferring drug resistance by computational analysis of the variants (**S1 Table**). Further, DNA sequencing established two mutations, i.e., T477A and T592S in *Tg*PRS, which substantially reduced susceptibility to L35 compared to wild-type parasites (**Fig 4**). Interestingly, both mutations are proximal to the two terminals of the PPL scaffold and not directly in the ATP, tRNA, or L-pro sites (**Fig 4**). This observation indicates that both the methyl-pyrazole terminal and hydroxy-methyl moieties may play a crucial role in the selectivity of L35. As for thermal stability, the *Tg*PRS-L35 and *Hs*PRS-L35 had a higher ΔTm value than the other four PPL derivatives. In addition, the thermal stability was higher with L-pro and the compound combination with ATP, or L-pro + ATP (**Figs 2 and 5**). While the possibility of off-site targeting by ATP-mimetics cannot be dismissed, genetic selection data shown here provide evidence these compounds target only PRS. These ATP mimetics and their stereoisomers (e.g., L35 and L36), while occupying the same catalytic pocket, exhibit specific and distinct inhibition profiles. We recently showed that HFG and L95 can indeed bind together to *Tg*PRS, covering all three of the enzyme-substrate subsites [49]. HFG and L95 compounds act as a triple-site inhibitor set forming an unusual ternary complex wherein HFG occupies the 3'-end of tRNA and the L-pro binding sites and L95 occupies the ATP pocket. These data provide an avenue for structure-based activity enhancement of this chemical series as anti-infectives. Further characterization of the PPL derivatives described in this study will support drug tailoring and development against toxoplasma and related parasites.

## Supporting information

**S1 Fig. Activity of MMV drugs on *Toxoplasma gondii* parasite and human cells. A-C)** Graphs showing the dose-response curves of *T. gondii* parasites in the presence of the MMV compounds indicated. Confluent HFFs were infected with wild-type (A), T447A (B) or T592S (C) edited parasites expressing the Nanoluc luciferase. After 48h of incubation, parasite proliferation was quantified to calculate the IC50 by non-linear regression analysis. The graphs represent the mean ±SD of 3 technical replicates from one experiment. Shaded error envelopes depict 95% confidence intervals. **D-E)** Dose-response curves of ARPE-19 and MDA231 cell lines in presence of different MMV drugs. Human cells were plated on 96 wells plates and incubated with growing concentrations of drugs. After 72h, the cells' viability was revealed using the CellTiter-Blue assay kit (Promega) and the $CC_{50}$ was calculated. The graphs represent the mean ± SD of 3 technical replicates from one experiment.
(TIF)

**S2 Fig. Activity of MMV35 on PRS mutant parasites. A)** Schematic of *PRS* gene editing strategy to introduce point mutations in *T. gondii* parasites. Focus on the *PRS* locus and CRISPR/Cas9-mediated homology-directed repair with single-stranded oligo DNA nucleotides (ssODNs) carrying nucleotide substitutions (red letters). After homologous recombination (HR) events, PRS recombinant parasites were selected with L35. Only T592S is shown for clarity. **B)** Images of plaques formed by recombinant parasites after 7 days in presence of 0.1μM of L35 or 0.1%DMSO. **C)** Dose-response curve of T477A and T592S edited parasites incubated with increasing concentration of Pyrimethamine or L35. The graphs represent one of three

different experiments. The mean ±SD is from 3 technical replicates of one assay and the shaded error envelopes depict 95% confidence intervals.
(TIF)

**S3 Fig. _Tg_PRS active site showing structural interactions associated with the five PPL-derivatives.** The structural interactions associated with **(A)** L95, **(B)** L96, **(C)** L97, **(D)** L35, **(E)** L36 in _Tg_PRS active site are shown.
(TIF)

**S4 Fig. _Hs_PRS active site showing structural interactions associated with the three PPL-derivatives.** The structural interactions associated with **(A)** L95, **(B)** L96, **(C)** L97 in _Hs_PRS active site are shown.
(TIF)

**S1 Table. Mutations found in candidate genes by RNA-Sequencing analysis of L35-resistant mutants.** The varying alleles of the resistant mutants found in candidate gene PRS are displayed. Amino-acid substitutions with the corresponding codons shown in parentheses are indicated for each mutagenized _T.gondii_ L35-resistant strain.
(DOC)

**S2 Table. Growth conditions for PRS crystals.**
(DOC)

**S3 Table. Summary of PDB entries associated with this manuscript.**
(DOC)

**S4 Table. Summary of data collection and refinement statistics.** Statistics for the highest-resolution shell are shown in parenthesis.
(DOC)

**S5 Table. Summary of $EC_{50}$ values of WT and CRISPR-Cas9 SDM-induced mutants against L35.**
(DOC)

**S1 Data. RNA-seq, sequence alignment and variant calling data files.**
(XLSX)

## Acknowledgments

MMV and Takeda Pharmaceutical Company Limited provided the ATP binding pocket inhibitor compounds used in the present study. We thank scientists in PROIXMA 1 and PROXIMA 2A beamlines, SOLEIL Synchrotron, France for the help in data collection. We are grateful for beamtime at the Diamond Light Source (DLS) and the staff of beamline I04 and I24 for data collection (BAG application mx14744).

## Author Contributions

**Conceptualization:** Mohamed-Ali Hakimi, Amit Sharma.

**Data curation:** Manickam Yogavel, Siddhartha Mishra, Nipun Malhotra, Jyoti Chhibber-Goel, Karl Harlos.

**Formal analysis:** Manickam Yogavel, Alexandre Bougdour, Siddhartha Mishra, Jyoti Chhibber-Goel.

**Funding acquisition:** Benoît Laleu, Mohamed-Ali Hakimi, Amit Sharma.

**Investigation:** Alexandre Bougdour, Siddhartha Mishra, Nipun Malhotra, Valeria Bellini.

**Methodology:** Alexandre Bougdour, Siddhartha Mishra, Nipun Malhotra, Valeria Bellini.

**Project administration:** Manickam Yogavel, Benoît Laleu, Mohamed-Ali Hakimi, Amit Sharma.

**Resources:** Karl Harlos, Benoît Laleu.

**Supervision:** Mohamed-Ali Hakimi, Amit Sharma.

**Validation:** Manickam Yogavel, Siddhartha Mishra.

**Visualization:** Alexandre Bougdour, Siddhartha Mishra, Jyoti Chhibber-Goel, Valeria Bellini.

**Writing – original draft:** Jyoti Chhibber-Goel.

**Writing – review & editing:** Manickam Yogavel, Alexandre Bougdour, Siddhartha Mishra, Nipun Malhotra, Jyoti Chhibber-Goel, Mohamed-Ali Hakimi, Amit Sharma.

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
