## [Decision Letter · Decision Letter 0]

23 Oct 2022

Dear Dr. Sharma,

Thank you very much for submitting your manuscript "Targeting prolyl-tRNA synthetase via a series of ATP-mimetics to accelerate drug discovery against toxoplasmosis" for consideration at PLOS Pathogens. As with all papers reviewed by the journal, your manuscript was reviewed by members of the editorial board and by several independent reviewers. The reviewers appreciated the attention to an important topic. Based on the reviews, we are likely to accept this manuscript for publication, providing that you modify the manuscript according to the review recommendations.

Sincerely,

Laura J. Knoll

Pearls Editor

PLOS Pathogens

Kami Kim

Section Editor

PLOS Pathogens

Kasturi Haldar

Editor-in-Chief

PLOS Pathogens

orcid.org/0000-0001-5065-158X

Michael Malim

Editor-in-Chief

PLOS Pathogens

orcid.org/0000-0002-7699-2064

Reviewer's Responses to Questions

**Part I - Summary**

Reviewer #1: In the study by Yogavel et al., a thorough inhibitor study on Toxoplasma gondii and human prolyl-tRNA synthetase (PRS) was performed, using an interdisciplinary approach, involving chemical mutagenesis and reverse genetics, biochemical and structural analysis. PRS has been previously described as a promising drug target and similar biochemical studies (using different inhibitors) have been previously described by the same groups. Here, 5 compounds based on the ATP-mimetics 1-(pyridin-4-yl) pyrrolidin-2-one (PPL), named L95, L96, L97, L35 and L36 were investigated in great detail.

Briefly, the authors found that L35 shows the highest selective inhibition by comparing toxicity on the parasite and its host cells. Next, chemical mutagenesis was performed and 2 mutants (T477A and T592A) in the coding region of TgPRS were identified to confer resistance to L35 and other PPL derivatives, clearly demonstrating that PRS is the target.

Using well established biochemical assays, inhibition of human and TgPRS was compared using these inhibitors and the in vitro data nicely reflect the findings in vivo.

Finally, the authors succeed in solving and comparing several structures of human and parasite PRS bound to inhibitors and L-Pro.

In summary, this well performed study opens an avenue for future structure-based optimisation of PRS inhibitors that appear to be very promising for future intervention strategies.

Since this reviewer has no significantl experience in structural biology, minor concerns are only on the remaining experiments.

Reviewer #2: This is a comprehensive study of a series of Toxoplasma gondii prolyl-tRNA synthetase (PRS) inhibitors based on the 1-(pyridine-4-yl) pyrrolidin-2-one (PPL) scaffold that targets the ATP binding pocket of the PRS enzyme. The work is thorough and well thought out, clearly demonstrating these inhibitors are targeting the Tg PRS (through generation of resistant mutants) and comparison of the binding of these inhibitors to human vs parasite PRS by thermal shift binding assays and crystallization studies. The work is important and adds considerable information to further the development of PRS inhibitors for anti-apicomplexan drug development. However this reviewer has some concerns with the narrowness of the selectivity indices, the lack of CC50 data on the host cells the parasites are grown in, and the validity of EC50 values that are close to or exceed the CC50 values in the case of resistant mutants. Overall, the manuscript could also use some careful grammatical editing, and the authors should confirm that statements made in the text match the data shown in the figures.

Reviewer #3: Summary:

In this manuscript, the authors provide evidence that five ATP-mimetic compounds developed to bind human and Plasmodium prolyl-tRNA synthetase (PRS) also bind PRS from Toxoplasma gondii (Tg) and inhibit its enzymatic activity and parasitic growth. The authors show these compounds inhibit Tg growth in HFFs and have approximately 1-10-fold selectivity by cytotoxicity in an epithelial and breast cell line and up to a 14x lower EC50 than the standard of care Tg drug pyrimethamine. They used a chemical mutagenesis screen to select for parasites resistant to the compound with the lowest EC50, L35, and discovered two mutations of interest in the PRS coding region, T477A and T592S. They confirmed at least one of these mutations confer resistance of the parasite to all five compounds, though which mutants are resistant to which compounds varied. The authors then proceed to characterize the biochemical inhibition of both the Tg and human PRS enzymes with each of these compounds and found that 3 are substantially more selective for the Tg PRS than human. The enzymatic data were confirmed with thermal shift data in the presence and absence of PRS substrates. Finally, the authors supplied excellent high resolution x-ray crystal structures of all five inhibitors bound to Tg PRS and three bound to human PRS. These structures revealed a consistent core binding mode of the PPL scaffold with specific nuanced differences in binding between the different compounds which provide structural evidence for differences in selectivity and activity. The authors conclude that these compounds are potential anti-infectives for Tg disease and the structural and biochemical data form a solid backbone for structure-guided development of these compounds as novel Tg drugs.

Impressions:

This paper provides a strong argument for these PPL compounds to be further developed as anti-infectives for Tg. The compounds already show selectivity over human PRS and reasonable biological selectivity with cell lines. The emergence of biological mutants with resistance is always a concern with any new drug development; however, the authors have identified two critical mutants at this early stage which provides important testing points for new derivatives. A weakness is the lack of biochemical and structural evaluations of these or other mutants, which would be useful to cross-validate the study conclusions here.

The crystal structures are a major strength of this manuscript. All structures are of high resolution with all but one greater than 2.5 A and 6/14 less than 2 A. Model fitting statistics suggest good to excellent agreement in the model to the experimental data and ligand fit was performed diligently. Minor issues are present with data presentation and numbering/unit clarity and consistency, and these are detailed below. All but one structure was available in the PDB for review and at least one was previously deposited and included in a different publication by the same group. The crystallographic data was included for easy comparison and appreciated.

The authors provide compelling data to explain selectivity and activity differences. The discussion succinctly summarizes these findings, though could be expanded to discuss other resistance mechanisms, such as relevance of the adaptive proline response mentioned in the introduction, and whether novel derivatives would need to avoid the proline binding site to avoid these mechanisms.

Overall, this paper represents an important step in identifying new compounds to develop into therapies for Tg infections, which is a major unmet need in infectious disease pharmacology. The structural, biochemical, and biological data presented here will not only be useful in further development but may have applications for other parasites, such as Plasmodium, and may become a new class of therapeutic inhibitors.

**Part II – Major Issues: Key Experiments Required for Acceptance**

Reviewer #1: This reviewer is unable to comment on the structural analysis, since it is far outside of his expertise. All other experiments have been performed to the highest standard and there are no major concerns.

Reviewer #2: Introduction/Fig 1: It is stated that these inhibitors were developed to target HsPRS as anti-fibrosis drugs. Were these modified in any way to optimize the interaction with TgPRS or are these the inhibitors developed to target HsPRS, and were identified by screening for activity against Tg or TgPRS? The author also mention that PfPRS was also targeted with PPL derivatives. Are these the derivatives described here as having activity against TgPRS? Please clarify.

Fig 2 C and D-the selectivity indices overall are very narrow for these PRS inhibitors. Because of this, it would be best if the CC50 and SI for the HFF host cells was also provided. It is not sufficient to just visually examine the host cells (line 228).

In Fig 2: L36 has an EC50 of 5800 nM (Fig 2A). The CC50 for this compound was never evaluated (Fig 2C and D; and Fig 4E). Is the compound specific for the parasite or is it killing the parasite via effects on the host cell?

Lines 255-256: “Notably, the mutation T477A significantly impaired the growth of L35-treated parasites (Fig. 3D lower panel, S2C) compared to pyrimethamine-treated parasites” It is unclear what comparison is being made here, or what point the authors are trying to make.

Fig 4E. It is unclear how the EC50s were determined as, in some cases, the EC50 exceeds the CC50 or is very close to the CC50 (For example L95 against T592S and L97 against T477A). Surely inhibition of host cells would affect growth of the parasite and confound calculation of the EC50?

Also in Fig 4E, the T592S mutation made the parasite considerably more susceptible to derivative L36. Perhaps the authors could comment on this?

Reviewer #3: This study identified two PRS mutants with reduced biological susceptibility to L35. Biological resistance of these mutants was validated by complimenting wt Tg with the same mutants which confirmed the phenotype. The mechanism by which these mutants reduce susceptibility was not validated. An evaluation of the biochemical activity of these compounds with purified PRS T477A and PRS T592S would be useful to suggest whether these mutations reduce biological L35 activity due to lower enzymatic inhibition/binding or via some other indirect mechanism.

The co-crystal structures are of high resolution and the Fo-Fc difference maps suggest accurate placement of the ligands into the electron density, though the lowest resolution structure (7FAL) has less connectivity for assignment. It would be useful to have a biochemical validation of these inhibitor models by mutating a residue that the model shows interacts with the inhibitors (such as but not limited to T482) and observing any reduced binding affinity/activity biochemically. Understandably any active site mutagenesis may impact enzymatic activity, so if selecting a mutant that does not impact conformation/activity is not possible, computational predictions such as docking in a predicted mutant model may be a useful surrogate test to validate the placement of ligands in the model.

**Part III – Minor Issues: Editorial and Data Presentation Modifications**

Reviewer #1: 1.) line 255ff: “Notably, the mutation T477A significantly impaired the growth of L35-treated parasites (Fig. 3D lower panel, S2C) compared to pyrimethamine-treated parasites (Fig. 3D upper panel, S2C).”

T477A confers more resistance compared to T592S? In contrast (and as expected) treatment with pyrimethamine shows similar effect for WT and mutant parasites. Therefore, this sentence should read:

“Notably, the parasites with the T477A mutation significantly overcame inhibition of growth in presence of L35 (Fig. 3D lower panel, S2C) compared to pyrimethamine-treatment (Fig. 3D upper panel, S2C).

2.) Figure 2F not described in the text.

3.) Figure 4F is not described in the legend

Reviewer #2: Minor comments:

Figs 3D and Fig 4A-D would be better presented as tables, especially as the scales on each graph vary.

Fig 3 C vs Fig 3D. the green and orange colors which represent the mutants are reversed between the figures (T477A is orange in C and green in D).

Fig 4F lacks a legend.

Fig 5 A and B have no key to indicate which inhibitor is represented by which curve. Perhaps these figures would be better in the supplemental as the data is summarized in Fig 5C.

Fig 5 D and E: Thermal shift data shown in Fig 5D and E are described in lines 295-298, but the numbers in the text do not match the numbers in the table in Fig 5E

Reviewer #3: Line 27- abstract- unclear from sentence for which organism PRS is a validated drug target

Line 131- sequence length of Tg PRS is indicated but not human PRS…appears to be approx. 1015-1506 from deposited structures

Line 181- “concentrations greater than their Km” is a bit vague…consider indicating actual concentration used or scale factor from Km

Line 184-193- Please indicate the concentration of the protein for crystallization somewhere in the methods (or table S2 if all are different)

Line 190- were the drugs dissolved in DMSO before being added to the protein? If so, what is the final DMSO concentration or range?

Line 234 & Fig. 2A- values given in text are nanomolar and in figure are micromolar…readability would be improved with consistent units.

Line 326 & 328-results- Figs. 7 & 8 were called out before Fig. 6…consider reordering/renaming

Line 327 & Line 661 Fig. 6 caption- Text (327) indicates inhibitors were verified at 3 sigma, but the figure (661) shows OMIT contour maps at sigma. Please confirm whether OMIT maps are contoured at 1 sigma or 3 sigma.

Line 357 & Fig. S3- Interactions and H-bonds are labeled in panels A & B but not in C-E. Additionally, the text states that there is no H-bond with L36 and R481, but the distance between the guanidinium and the methylpyrazole of L35 (7FAN) is approx. 3.4A and the distance between the guanidinium and the methylpyrazole of L36 (7FAL) is approx. 3.3A…what criteria was used to determine H-bond existence? It could be helpful to show H-bond distances in Fig. S3 if space allows/visually possible.

Line 371 & Fig. 8B- “motif 1” indicated in text but labeled as “motif” in Fig. 8 panel B.

Line 379 & 380- please indicate which species’ residues are in parenthesis and which are not

Fig. 5A&B- which color/shape is which compound? Same as in panel D?

Fig. 5C- Units in text are nM but in figure are uM.

Supplemental Table S3-

1. The legend calls out Favored/Allowed/Outlier for the Ramachandran plot but only two numbers are shown in the boxes ex: 98/2. They appear to be Favored/Allowed only…please clarify the callout in the legend or add a decimal to indicate all 3, such as 97.5/2.1/0.4 since there is a small percentage (<0.5%) of outliers, and some (such as F479) are close to the inhibitor binding site and might be of interest to note if inhibitors influence rotamers.

2. The table could benefit from including additional information typical to the standard tables for reporting crystallographic data, and similar to Table 1 in the authors’ companion publication (Manickam, Y. et. al. 2022). Although adding all of the collection and refinement statistics in the table may be cumbersome, reporting the resolution as shells and highest shell would be useful, as well as including I/sigmaI, completeness, and RMS deviations.

3. All structures were available in the PDB for review except 7FAK was still on hold. Although it was not required for this review, as the structures presented here were all of excellent quality, and this specific inhibitor had a complimentary structure in a different space group, the authors should be aware of the hold in case there was a mistake in the release date. If not, no concerns.

PLOS authors have the option to publish the peer review history of their article (what does this mean?). If published, this will include your full peer review and any attached files.

Reviewer #1: No

Reviewer #2: No

Reviewer #3: No

Figure Files:

Data Requirements:

Reproducibility:

References:

---

## [Decision Letter · Decision Letter 1]

16 Jan 2023

Dear Dr. Sharma,

We are pleased to inform you that your manuscript 'Targeting prolyl-tRNA synthetase via a series of ATP-mimetics to accelerate drug discovery against toxoplasmosis' has been provisionally accepted for publication in PLOS Pathogens.

Before your manuscript can be formally accepted you will need to complete some formatting changes, which you will receive in a follow up email. A member of our team will be in touch with a set of requests. You will also see that there are suggestions from reviewer 3 below that are easy clarifications, so it would be great if you could add those.

Best regards,

Laura J. Knoll

Pearls Editor

PLOS Pathogens

Kami Kim

Section Editor

PLOS Pathogens

Kasturi Haldar

Editor-in-Chief

PLOS Pathogens

orcid.org/0000-0001-5065-158X

Michael Malim

Editor-in-Chief

PLOS Pathogens

orcid.org/0000-0002-7699-2064

Reviewer Comments (if any, and for reference):

Reviewer's Responses to Questions

**Part I - Summary**

Reviewer #1: The authors thoroughly addressed mine and the other reviewers’ comments. I have no further concerns and congratulate the authors to a very nice study.

Reviewer #2: From my forst review: This is a comprehensive study of a series of Toxoplasma gondii prolyl-

tRNA synthetase (PRS) inhibitors based on the 1-(pyridine-4-yl) pyrrolidin-2-one (PPL)

scaffold that targets the ATP binding pocket of the PRS enzyme. The work is thorough

and well thought out, clearly demonstrating these inhibitors are targeting the Tg PRS

(through generation of resistant mutants) and comparison of the binding of these

inhibitors to human vs parasite PRS by thermal shift binding assays and crystallization

studies. The work is important and adds considerable information to further the

development of PRS inhibitors for anti-apicomplexan drug development.

The authors have responded to all my concerns and comments satisfactorily. I have no further critiques

Reviewer #3: Summary of changes:

The authors were both appreciative of the reviewers’ positive comments on the strengths of the manuscript and carefully considered and responded to the reviewers’ criticisms. The authors have made changes to address all the reviewers’ minor points which encompassed clarity of data presentation, inclusion of additional details regarding data presented in the original submission, consistency in numbering and units, and grammatical and visual presentation issues. The authors did respond to major concerns with rational explanations, but no new experiments were conducted to address the same. Overall, the response was thorough, well thought out, and has improved the manuscript.

Impression & opinion:

All experiments and data presented herein are of excellent quality and rigor and the findings are significant. Addressing the minor concerns improved the paper considerably. Additional experiments requested by the reviewers in the “major concerns” sections would have strengthened the manuscript further; however, the results of these experiments would not have impacted the major conclusions of this study and the authors explanation as to why they were not done is acceptable. Recommend adding brief language in the discussion to those points and addressing a small number of minor concerns prior to publication.

Response to summary:

The authors clearly appreciate the attentiveness the reviewers made to this work and that is gladly noticed as well. The authors choice in cell line for toxicity tests is appropriate since they are analogous cells to those found in clinical infections and estimate selectivity indices more conservatively, which will be valuable in subsequent development. The authors correctly state that these compounds are not claimed as drug leads (nor does that claim appear in this or prior versions) but are hits discovered and characterized early in the process. Subsequent development will certainly focus on deriving compounds with improved SI.

**Part II – Major Issues: Key Experiments Required for Acceptance**

Reviewer #1: none

Reviewer #2: None

Reviewer #3: Response to major points:

The companion manuscript assessing activity in Plasmodium will be of great interest to the field. Thank you for including additional context in this manuscript.

Thank you for clarifying the limits of CC50 calculation with L36, for clarifying the plaque size comparison, for acknowledging that host cells could be affected with the EC50 experiments, and for the comment on L36 mutant susceptibility.

The related explanation of cell line choice for CC50 is acceptable, but please include brief language (similar to the response given) as to why HFFs were only examined visually, and these other cell lines were chosen for CC50.

The forward genetic approach is a powerful method for identifying resistance mutations and is a major strength of this work. In agreement with the authors, it is likely that the mechanism of resistance is via lowering the affinity of the target for these compounds, though reviewer 2 and the authors themselves do point out that the T592S mutant has increased affinity to L36, reversing this trend, likely due to enantiospecificity. Although additional experiments with recombinant protein for each mutation would be helpful to confirm the changes in target affinity (both positive and negative), it would not change the conclusions of the studies here, nor does lack of these experiments diminish the importance of the biological model presented here. Recombinant biochemical experiments will be helpful for mechanistic evaluations and new moiety design when testing derivatives of these compounds and should be considered for future work.

**Part III – Minor Issues: Editorial and Data Presentation Modifications**

Reviewer #1: none

Reviewer #2: none

Reviewer #3: Response to minor points:

The authors thoroughly addressed all reviewers’ minor points and are commended for the inclusion of expanded details in their supplementary tables. All responses were satisfactory, but some information from those responses did not make it to the resubmitted manuscript.

Specifically:

1) Please include the final DMSO concentration for crystallization either as a general range in the methods or specific % in the supplementary tables.

2) Please cite the PLIP server (pick appropriate citation for version on their website) since it was used to determine H-bond patterns.

PLOS authors have the option to publish the peer review history of their article (what does this mean?). If published, this will include your full peer review and any attached files.

Reviewer #1: No

Reviewer #2: No

Reviewer #3: No

---

## [Editor Report · Acceptance letter]

13 Feb 2023

Dear Dr. Sharma,

We are delighted to inform you that your manuscript, "Targeting prolyl-tRNA synthetase via a series of ATP-mimetics to accelerate drug discovery against toxoplasmosis," has been formally accepted for publication in PLOS Pathogens.

Best regards,

Kasturi Haldar

Editor-in-Chief

PLOS Pathogens

orcid.org/0000-0001-5065-158X

Michael Malim

Editor-in-Chief

PLOS Pathogens

orcid.org/0000-0002-7699-2064